# Ingestion of Soybean Sprouts Containing a HASPIN Inhibitor Improves Condition in a Mouse Model of Alzheimer’s Disease

**DOI:** 10.3390/biology12020320

**Published:** 2023-02-16

**Authors:** Hiromitsu Tanaka, Hiroaki Matsushita, Keizo Tokuhiro, Atsushi Fukunari, Yukio Ando

**Affiliations:** 1Faculty of Pharmaceutical Sciences, Nagasaki International University, 2825-7 Huis Ten Bosch, Sasebo 859-3298, Japan; 2Department of Genome Editing, Institute of Biomedical Science, Kansai Medical University, Hirakata City 573-1191, Japan

**Keywords:** HASPIN, coumestrol, kinase, tau, amyloid, soybean, 5xFAD

## Abstract

**Simple Summary:**

HASPIN plays important roles in chromosome segregation during cell division by phosphorylating histone H3 at threonine. We showed that HASPIN was expressed in the hippocampus and phosphorylated the tau protein, acting as a phosphorylation substrate. Oral administration of soybean sprouts grown for increased coumestrol (a HASPIN inhibitor) decreased the phosphorylated tau protein level in the hippocampus and suppressed short-term memory loss in the Alzheimer’s disease model mice (5xFAD). Functional analysis of HASPIN may help treatment of Alzheimer’s disease.

**Abstract:**

The MATP/tau protein is hyperphosphorylated in Alzheimer’s patients. Therefore, research into the regulation of tau protein phosphorylation is important for understanding Alzheimer’s disease. HASPIN is a serine/threonine kinase that is expressed in various cells. To examine whether HASPIN is involved in the onset of Alzheimer’s disease through tau protein phosphorylation, we investigated the effects of a diet including soybean sprouts rich in the HASPIN inhibitor coumestrol in a mouse model of Alzheimer’s disease (5xFAD). The results showed that HASPIN was expressed in the hippocampus and phosphorylated tau protein, while the ingestion of soybean sprouts containing coumestrol suppressed the development of spatial cognitive dysfunction in 5xFAD. These results indicate that HASPIN may be one of the target molecules for the repression of tau phosphorylation in the treatment of Alzheimer’s disease.

## 1. Introduction

*HASPIN* has been cloned as a gene encoding a serine/threonine kinase intensely expressed in haploid germ cells, and it was later found to be expressed at low levels in various tissues [1,2]. It has been reported that HASPIN is conserved in various eukaryotes [3,4] and phosphorylates histone H3 at threonine 3 [5]. HASPIN plays an important role in the function of the chromosome passenger complex and regulates chromosome segregation during cell division [6]. Functional analysis of HASPIN in germ cells revealed that it is involved in the germ-cell-specific phosphorylation of Thr127 of TH2A, which is involved in chromosome segregation during meiosis [7]. Furthermore, HASPIN has been found to be involved in the phosphorylation of the hyaluronic acid binding protein 1 (HABP1/p32/gC1qR/C1QBP), playing an important role in fertilization. It also binds to MORN2, which is involved in acrosome formation, as well as to t-complex TCP10, RANBP9, and so on, which are associated with microtubules [8]. These results collectively indicate that HASPIN plays multiple roles related to serine/threonine in spermatogenesis. Gene expression analysis in transgenic mice using a *HASPIN* promoter and *EGFP* as a reporter revealed that HASPIN was expressed in mature eggs, as well as in the inner cell mass of the blastocyst during post-fertilization embryonic development [9]. These results suggest that HASPIN is intensely expressed in proliferating cells, in addition to haploid sperm cells.

On the other hand, inhibitors of HASPIN have been identified and shown to specifically suppress the proliferation of various cancer cells [10]. Inhibition of HASPIN function is a major target for suppressing cancer cells, as cancer cell growth was also suppressed in RNA interference (RNAi) experiments against *HASPIN* [11]. In *HASPIN* gene-disrupted mice, the function of HASPIN was complemented by an unknown mechanism, and no significant defect was observed [12]. From this, the risk of side effects associated with *HASPIN* gene inhibitors is thought to be low. As a result of the functional analysis of HASPIN, regulation of the serine/threonine kinase activity of HASPIN is important for the cell division of normal cells, and its expression must be strictly regulated. As a disease involving abnormal cells that are out of autonomic control in vivo with aberrant phosphorylation of serine/threonine of cellular proteins, Alzheimer’s disease is well known as a cell disease associated with serine/threonine kinase. Alzheimer’s disease is caused by excessive phosphorylation of the tau/MATP protein, which accumulates in cells and disrupts cell functions [13]. In modern long-lived societies, the number of patients with cancer or Alzheimer’s disease has been increasing, which are diseases that should be resolved at an early stage. In this study, we identify that the tau protein is involved as a phosphorylation substrate of HASPIN, and examine the effects of HASPIN inhibition on Alzheimer’s disease in vivo and in vitro.

## 2. Materials and Methods

### 2.1. Animals

C57BL/6J mice were purchased from CLEA Japan (Tokyo, Japan) and sacrificed by cervical dislocation immediately prior to the experiments. All animal experiments conformed to the Guide for the Care and Use of Laboratory Animals and were approved by the Institutional Committee of Laboratory Animal Experimentation and Research Ethics Committee of Nagasaki International University (approval number 148). This research did not include any experiments with human subjects performed by any of the authors. The mice were maintained under specific pathogen-free conditions in the animal experimentation facility at Nagasaki International University, with temperature and lighting controlled throughout the experimental period. The mice were provided with food and water ad libitum. Hi-Durability IRRD M/R (Japan SLC, Shizuoka, Japan) was used as the diet for the mice. The sprout-containing diet (15 µg/g: coumestrol/diet) was prepared by mixing sprout powder and Hi-Durability IRRD M/R. Soybean sprouts were grown, according to previous papers, to contain approximately 100 µg/g of coumestrol [14]. The sprouts were dried at 75 °C overnight and powdered using a mixer. We moistened the mouse food (500N Hi-Durability IRRD M/R) in 50 °C hot water, added up to 15% sprout powder and 10% potato starch as a binding agent to the food, cut the food into dumplings with a diameter of 3 cm, and dried at 80 °C overnight. The mice were fed under each condition from 8 weeks of age and used in experiments at 5 months of age. Considering the average 4.5 g of diet per day consumed by a mouse and the amount of coumestrol contained in the diet, the mice consumed 67.5 µg of coumestrol per day, which was within the intake limits for isoflavones in humans relative to body weight.

### 2.2. In Vitro Kinase Assay

A total of 10 ng of recombinant human HASPIN (PV5708; Life Technologies Japan, Tokyo, Japan) and 1 µg of recombinant human tau (Tau-441 (AG960); Sigma-Aldrich, St. Louis, MO, USA) were mixed in the reaction solution (40 mM HEPES pH 7.4; 10 mM MgCl_2_; 3 mM MnCl_2_; 5 mM CaCl_2_; 150 mM NaCl). Then, 10 μCi γ32P-ATP was added to the reaction mixture and adjusted to 40 μL. The recombinant human HASPIN consists of the kinase domain of human HASPIN and a glutathione S-transferase as a tag, expressed using a baculovirus vector. Tau-441 human (recombinant from *E. coli*) is the tau isoform variant 2N4R, considered the largest human brain tau protein. The reaction was carried out at 37 °C for 30 min. After the reaction, a sample buffer for SDS-PAGE was added and centrifuged. Then, we applied SDS-PAGE. Inhibitors (5 µg/µL) were stored in 100% DMSO at −30 °C and added to the reaction mixture to achieve each concentration. The signals were observed using a STORM 820 Phosphorimager (GE Healthcare life sciences). The Image processing program ImageQuant TL (Amersham, MA, USA) was used to analyze the intensity of the signal.

### 2.3. Hippocampal Slice Experiment

The mice were decapitated, and their brains were rapidly removed and placed into ice-cold artificial cerebrospinal fluid (ACSF) containing the following: 124 mM NaCl, 4.4 mM KCl, 2.5 mM CaCl_2_, 1.3 mM MgSO_4_, 1 mM NaH_2_PO_4_, 26 mM NaHCO_3_, and 10 mM glucose. The hippocampi were extracted, and transverse hippocampal slices (500 μm thickness) were cut and placed into ACSF medium microtube and left to stand for 30 min at room temperature. The hippocampal slices were transferred to an ACSF medium and incubated for 30 min. CHR-6494 (5 µg/µL) was stored in 100% dimethyl sulfoxide (DMSO) at −30 °C and added to each medium to achieve each concentration. The sample of hippocampal slices was subjected to Western blot analysis.

### 2.4. Western Blot Analysis

First, 10–50 μg of tissue or cultured cell proteins was separated by SDS-PAGE. Then, it was electroblotted onto PVDF membranes. After blocking with 4% skimmed milk in TBS-T (50 mM Tris-HCl, pH 7.4; 138 mM NaCl; 2.7 mM KCl; 0.1% Tween 20), the membranes were reacted with primary antibodies overnight at 4 °C and then with secondary antibodies for 90 min at room temperature. The antigen–antibody complexes were detected using ECL Select (Cytiva Amersham, MA, USA). The signals were observed using an Amersham Imager 680 (Cytiva Amersham), Image Quant LAS4000 (Fujifilm, Tokyo, Japan), or POD Immunostain Kit (Wako, Osaka, Japan). The antibodies used in the experiment are detailed in Table 1. The Image processing programs ImageJ and ImageQuant TL were used to analyze the signal intensity [15].

### 2.5. Reverse Transcriptase-Polymerase Chain Reaction (RT-PCR)

Total RNA from mouse organs was extracted as per the protocol of ISOGENII (NIPPON GENE, Tokyo, Japan). RNA was extracted from three male mice bred under each condition. Then, 1 μg of total RNA was reverse transcribed using the oligo dT primer and PrimeScript RT reagent Kit (Takara Bio, Tokyo, Japan), according to the manufacturer’s recommendations. PCR analysis of samples was performed using EmeraldAmp MAX PCR Master Mix (Takara Bio). *HASPIN* and *Actb*/β-*ACTIN* were amplified using the upper primer for *HASPIN* (5′-TTGAGCACCGGGACTTACAC-3′), the upper primer for Actb (5′-CCGTGAAAAGATGACCCAGAT-3′), the reverse primer for *HASPIN* (5′-CCATTGAGGGTGTAGCGGAG-3′), and the reverse primer for *Actb* (5′-GTACATGGCTGGGGTGTTG-3′). The following PCR conditions were used: 32 cycles for *HASPIN* and 28 cycles for *Actb* of denaturation at 94 °C for 30 s, annealing at 58 °C for 30 s, and extension at 72 °C for 15 s. The PCR products (*HASPIN*: 79 bp and *Actb*: 55 bp) were electrophoresed in 2% agarose and detected with ethidium bromide.

### 2.6. Transfection

The HT-22 mouse hippocampal neuronal cell line (Merck, Darmstadt, Germany) was cultured in Dulbecco’s modified Eagle’s medium (DMEM) containing 10% fetal bovine serum and 100 units/mL of penicillin–streptomycin (Nacalai, Kyoto, Japan) [16]. Transfection was performed according to the manufacturer’s recommendations, using Lipofectamine 3000 (Invitrogen, Waltham, MA, USA). HT-22 cells were transfected with pEGFP-C1 expression vectors (CLONTECH, Mountain View, CA, USA) containing *egfp*, *egfp–HASPIN*, or *egfp*–mutant *HASPIN* [1]. In these expression vectors, *HASPIN*s were full-length cloned into the multiple cloning site of the pEGFP vector. The cells were harvested after culturing for 24 h.

### 2.7. Microscopic Observations

For morphological observation, the brains were fixed in 4% paraformaldehyde, embedded in paraffin, and sectioned at a thickness of 10 µm. Deparaffinized sections were stained with hematoxylin and eosin and observed using a BX50 microscope (OLYMPUS, Tokyo, Japan). The cultured cells were observed using an IX70 fluorescence inverted microscope (OLYMPUS).

### 2.8. Congo Red Staining

Paraffin sections of mouse brain were stained in Congo red solution (Fujifilm, Tokyo, Japan) for 60 min. Then, the sections were stained in hematoxylin (Congo red, Fujifilm, Tokyo, Japan) for 1 min, washed twice in 100% ethanol, and mounted using NEW MX reagent (Matsunami Glass Industries, Osaka, Japan).

### 2.9. Y-Maze Test

A Y-maze test was performed to examine spontaneous locomotor activity and spatial working memory [17]. A mouse was placed at the end of one arm of a device, in which three arms of the same size (assumed to be A, B, and C) were connected at 120°, then allowed to move freely for 8 min, and their behavior was recorded. We recorded the number of times each arm was entered. Spontaneous alternation is the behavior of exploring novel environments. In this test, we analyzed the rate at which mice enter different arms consecutively. If they entered the same arm many times, we judged that they did not remember their own exploratory behavior; that is, their spatial working memory ability was low. The spontaneous locomotor activity was calculated from the total number of arm entries. Mice were given a diet containing soybean sprouts from 8 weeks old, and the experiment was conducted at 5 months old. Ten mice were used for each condition and statistically processed.

### 2.10. Statistical Analysis

All data were derived from at least three independent experiments. The results are expressed as means ± standard deviation (SD), or standard error (SE) for each condition. Differences between the experimental and control conditions were determined using the two-tailed paired *t*-test, and a *p*-value of <0.05 was considered statistically significant. Tukey–Kramer’s post hoc analysis was used to compare multiple conditions, and a *p*-value of <0.05 was considered statistically significant.

## 3. Results

### 3.1. Phosphorylation of the Tau Protein by HASPIN

HASPIN is a serine/threonine kinase. We sought to examine whether the recombinant tau protein was phosphorylated by the recombinant HASPIN. The recombinant human tau protein was incubated with recombinant human HASPIN and, as a result, the signals of the phosphorylated tau protein were obtained (Figure 1). These signals were decreased at an IC50 of approximately 5 nM and 15 µM with the addition of a HASPIN inhibitor CHR-6494 (ChemScene, Monmouth Junction, NJ, USA) and coumestrol (Sigma-Aldrich, MO, USA), respectively, as in previous reports [18,19]. These results indicated that the kinase activity of HASPIN directly phosphorylated the tau protein.

### 3.2. Expression of HASPIN in the Hippocampus

*HASPIN* is expressed in various organs but was not detected in the brain by Northern blot analysis [2]. First, we examined *HASPIN* expression in the hippocampus by RT-PCR. As a result, the expression of *HASPIN* was observed in the hippocampus (Figure 2). Next, we examined the expression of *HASPIN* in Alzheimer’s disease model mice, which are transgenic mice expressing transgenes with five familial Alzheimer’s disease mutations (5xFAD) in the amyloid precursor protein and Presenilin1 [17]. The expression of *HASPIN* was found to be higher in 5xFAD male mice fed standard food (1.27 ± 0.01) and the food including soybean sprouts, compared to that in wild-type mice (1.39 ± 0.03). These data indicate that the expression level of *HASPIN* mRNA in the 5xFAD mice was higher than that of the wild-type, and the diet supplemented with soybean sprouts containing a HASPIN inhibitor did not reduce *HASPIN* mRNA expression in 5xFAD mice.

### 3.3. The Phosphorylation of Tau in the Hippocampal Slices

We examined whether tau phosphorylation was inhibited in the hippocampal slices when using the HASPIN inhibitor CHR-6494. Western blot analysis using phosphorylated antibodies showed that the addition of inhibitors decreased the phosphorylation of the tau protein with 20 (0.46 ± 0.04) or 100 nM (0.25 ± 0.11), compared with the control (0.80 ± 0.14); see Figure 3.

### 3.4. The Phosphorylation of Tau Protein in HT-22 Cells

We examined the effect on tau protein phosphorylation when overexpressing recombinant HASPIN in the HT-22 cultured cells of mouse hippocampal neuronal cells. Western blot analysis indicated that the amount of phosphorylated tau protein increased approximately 5 times in the HT-22 cultured cells expressing HASPIN (1.84 ± 0.1), compared with those expressing EGFP (0.37 ± 0.1); see Figure 4. Mutant HASPIN (1.51 ± 0.1), which lost kinase activity, also increased phosphorylated tau protein by about 4-fold. HASPIN and mutant HASPIN had localized to the nucleus at 24 h after transfection, and the cultured cells formed to round with the expression of the recombinant proteins but did not die (Figure 5). Wild-type and mutant HASPIN localized predominantly to the nuclei in a punctate manner. These results suggested that HASPINs, regardless of their kinase activity, may affect the microtubules that support cell morphology.

### 3.5. Effect of the Diet Containing Soybean Sprouts on Tau Protein Phosphorylation in Mice

Soybean sprouts contain the highest amount of the HASPIN inhibitor, coumestrol, among vegetables [14]. We investigated the effect of a coumestrol-enriched diet using soybean sprouts on tau protein phosphorylation in the hippocampus. The 5xFDA mouse is an Alzheimer’s model mouse that expresses amyloid β abundantly. Phosphorylated tau protein has been shown to be increased in 5xFAD mice compared with wild-type mice [20]. Therefore, we used 5xFAD mice to investigate the influence of HASPIN on changes in tau phosphorylation caused by amyloid β accumulation. Quantitation by luminescence using HRP as a secondary antibody showed that the value of intensity relative to β-actin in six male and six female mice fed a standard diet was 0.75 ± 0.35 (the mean ± SD), while the value of intensity relative to β-actin in six male and six female mice fed the diet including soybean sprouts was 0.33 ± 0.14. Both the wild-type and 5xFAD mice orally fed a diet including soybean sprouts showed significantly decreased phosphorylated tau protein, with a 0.22 times reduction (*p* = 0.004; Figure 6). No differences in results between male, female, wild-type, and 5xFAD mice were observed in these experiments. The diet formulation and average daily mouse food consumption (4–5 g) led to an intake of about 68 µg of coumestrol per day, equivalent to 135 mg per day for humans when converting to average human body weight (60 kg).

### 3.6. Effect of the Diet Containing Soybean Sprouts on Amyloid β Protein in Mice

It has been reported that coumestrol serves as a scavenger of reactive oxygen species (ROS) and can reduce the amount of amyloid β in the brain [21,22]. We examined the amount of amyloid β protein in 5xFAD mice orally fed soybean sprouts. Quantitation by luminescence using HRP as a secondary antibody showed that the value of intensity relative to β-actin in three male and three female 5xFAD mice fed a standard diet was 0.016 ± 0.002 (SD), while the value of intensity relative to β-actin in three male and three female 5xFAD mice fed a diet including soybean sprouts was 0.009 ± 0.002 (SD). The intensity of amyloid β protein in 5xFAD mice fed a diet including soybean sprout was 0.55 times lower than that in the 5xFAD mice fed a standard diet (*p* = 0.005; Figure 6). No differences in results between males and females were observed in these experiments. Congo red staining of the hippocampal tissue from the 5xFAD mice sections showed a significant reduction in the number of amyloid plaques in the group fed a diet including soybean sprouts (3.3 ± 1.1) when compared with the standard diet group (6.6 ± 0.5). Congo red staining of hippocampal tissue sections showed a significant reduction in the signal number of amyloid β aggregations (Figure 7). Overall, the results revealed that the 5xFAD mice fed a diet including soybean sprouts presented decreased amyloid β in the hippocampus.

### 3.7. Behavioral Experiment of the Mice Fed with Soybean Sprouts

It has been reported that 6-month-old mice on a normal diet showed short-term memory attenuation in a Y-maze [17]. Behavioral abnormalities in 5xFAD mice fed with soybean sprouts were examined in a Y-maze experiment. In this experiment, the short-term memory values of male and female 5xFAD mice fed a standard diet were 54.8 ± 8.6 and 55.2 ± 8.4, respectively. In addition, the short-term memory value of the male wild-type mice was 66.1 ± 7.1 and that of the female wild-type mice was 65.9 ± 9.3; therefore, the short-term memory values of both the male and female 5xFAD mice were significantly lower (Figure 8). On the other hand, despite the low values of the short-term memory for both male and female 5xFAD mice in the Y-maze experiment, the values for the 5xFAD male and female mice fed soybean sprouts were 66.5 ± 8.0 and 67.6 ± 10.5, respectively, showing no significant difference from the normal mice (Figure 8). These results indicated that the development of 5xFAD Alzheimer’s disease was suppressed in male and female mice that orally ingested soybeans, potentially due to their rich content of the HASPIN inhibitor coumestrol.

## 4. Discussion

It is known that HASPIN is expressed in trace amounts in various organs, and it plays important roles in chromosome segregation during cell division by phosphorylating histone H3 at threonine 3 [5]. The aberrant expression of HASPIN, which is tightly epigenetically regulated, appears to be associated with the induction of deautonomous and aberrant cells in vivo.

Alzheimer’s disease is caused by abnormal cells that are out of autonomic control in vivo, involving aberrant phosphorylation of serine/threonine of cellular proteins. In Alzheimer’s disease, pathological changes, such as senile plaques, neurofibrillary tangles, and neuronal cell death, are typically observed in the brain. Neuritic plaques consist of small peptides called amyloid β [13]. Neurofibrillary tangles are tau protein aggregates that form insoluble fibrous aggregates and accumulate within nerve cells. Tau is a protein involved in the stability of microtubules. Excessive phosphorylation of the tau/MATP protein prevents it from binding to microtubules, which is thought to destabilize the microtubules and reduce cell function. GSK3 and CDK5 are known to phosphorylate some sites of tau protein, leading to a decrease in its ability to bind to microtubules [23]. Among the pathological changes, senile plaques are the earliest to appear in the brain, followed by neurofibrillary tangles, brain atrophy (neuronal cell death), and then dementia. Analysis of the molecular mechanisms underlying phosphorylation of the tau protein is important for understanding the pathogenesis of Alzheimer’s disease.

In this study, we showed that HASPIN was expressed in the hippocampus and phosphorylated the tau protein, acting as a phosphorylation substrate. CHR6494 and coumestrol inhibited the phosphorylation of HASPIN, thus reducing tau protein phosphorylation. In hippocampal sliced cultured cells, the phosphorylated tau protein remained even after the HASPIN inhibitors were effective. In the experiments in which HASPINs were expressed in HT-22 cells, HASPIN expression increased the phosphorylated tau protein level. Expression of mutant HASPIN, which has lost its kinase activity, also increased phosphorylated tau protein for 80% of wild-type HASPIN. The mutant HASPIN showed similar subcellular localization as wild-type HASPIN. This indicates that the mutant HASPIN, which lacks kinase activity, retains some of its functions other than kinase activity. From these results, HASPIN may play a role as part of a protein complex that induces the phosphorylation of tau protein (e.g., by GSK3, CDK5, and so on). HASPIN may also be involved in the site-specific phosphorylation of tau protein, as it can be phosphorylated at multiple sites.

Oral administration of soybean sprouts grown for increased coumestrol (a HASPIN inhibitor) decreased the phosphorylated tau protein level in the hippocampus and suppressed short-term memory loss in the Alzheimer’s disease model mice (5xFAD). In this experiment, 5xFAD mice continuously orally ingested soybean sprouts (approximately 68 µg/day). Coumestrol inhibits HASPIN, but the IC50 concentration for inhibition was very high, and the amount of coumestrol in soybean sprouts does not appear to be sufficient to completely inhibit HASPIN.

Multiple pharmacological actions have been reported for coumestrol, which is a phytoestrogen known to act on female hormone receptors. Coumestrol acts on estrogen receptors with lower affinity than estrogen and also inhibits aromatase, 3β-hydroxysteroid dehydrogenase, and 17β-hydroxysteroid dehydrogenase (17β-HSD) [24,25]. As a result, it affects the regulation of steroid hormone production. Furthermore, coumestrol has the capacity to act as a scavenger of reactive oxygen species (ROS) [26] and also has an affinity for cholinesterase, the expression of which is suppressed by amyloid β [27]. Coumestrol also up-regulates Sirtuin-1 (SIRT1), which is an NAD-dependent deacetylase that participates in the regulation of cell senescence, metabolism, inflammation, and mitochondrial function [28]. It has recently been shown that coumestrol acts on monoamine oxidase (MAO) and inhibits the accumulation of amyloid β [22]. Soybean sprouts also contain some bioactive substances [14]. We found that the oral administration of soybean sprouts reduced tangled amyloid β, suppressed phosphorylation of the tau protein, and improved short-term memory in mice. The suppression effect of Alzheimer’s symptoms may appear through synergistic effects, in addition to the inhibition of HASPIN by coumestrol.

Oral administration of soybean sprouts grown for increased coumestrol to wild-type mice suppressed phosphorylation of the tau protein, but no abnormalities were observed in the spontaneous alternation experiment, and no significant behavioral abnormalities were observed. Detailed analysis in higher animals is required, but it is possible that coumestrol has no side effects.

As coumestrol is a phytoestrogen, we compared the effects of soybean sprout intake for males and females in the experiment, but no significant differences were observed in the effects on Alzheimer’s disease. It was reported that coumestrol treatment reduced adiposity and had antiobesogenic effects [29]. Male mice on a diet containing bean sprouts significantly lost weight; their weight was reduced to 83% of that of mice fed the standard diet (Appendix A). On the other hand, female mice on a diet containing bean sprouts lost weight, but the difference was not significant. The difference in the results between males and females may be due to the effects of coumestrol as a phytoestrogen. Females, with high estrogen at basal levels compared with males, may need higher phytoestrogen levels to induce supernumerary biogenic effects than males.

The function of the phosphorylated tau protein is unknown, and it will be interesting to analyze which parts of the tau protein are phosphorylated by HASPIN and its function in neurons, as well as its relationship with other serine–threonine kinases. In this paper, we showed that HASPIN is expressed in the hippocampus and phosphorylates tau protein and that soybean sprouts containing a HASPIN inhibitor suppress the onset of Alzheimer’s disease.

In this study, a diet containing 15% soybean sprout powder was found to be effective for weight loss in 5xFAD mice. However, considering the difference in body weight between mice and humans, the average human would have to eat 1 kg or more of bean sprout powder to consume a dose equivalent to that administered to the mice in this study. Such an amount is not realistic to achieve in the human diet. In addition, since soybean sprouts contain large amounts of water, it would be difficult for humans to ingest the required amounts of raw soybean sprouts. Although this experiment has its limitations, it will facilitate detailed analysis of the pathology, such as experiments with other Alzheimer’s model mice and animals other than mice. Furthermore, a cultivation method that increases the coumestrol content of soybean sprouts could make it feasible for humans to consume the required dose of coumestrol by eating these sprouts in more realistic quantities. The dose of coumestrol consumed by the mice in this study would equate to about 110 mg/day for the average human (assuming an average human body weight of 50 kg). The identification of the components contained in soybean sprouts and the elucidation of their molecular functions should advance the understanding of Alzheimer’s disease and the development of treatment methods.

## 5. Conclusions

HASPIN phosphorylates tau protein as a substrate. As such, CHR6494 and coumestrol—as HASPIN inhibitors—were found to inhibit tau phosphorylation. Furthermore, oral administration of coumestrol-rich soybean sprouts decreased the amount of amyloid β and phosphorylated tau in the brains of mice, and suppressed short-term memory decline in 5xFAD, an Alzheimer’s disease model mouse. Prevention of Alzheimer’s disease is an important issue, considering the progressive aging of modern societies. HASPIN may be involved in Alzheimer’s, so soybean sprouts containing HASPIN inhibitors are considered to be very promising ingredients for the prevention of Alzheimer’s disease.

## Figures and Tables

**Figure 1 biology-12-00320-f001:**
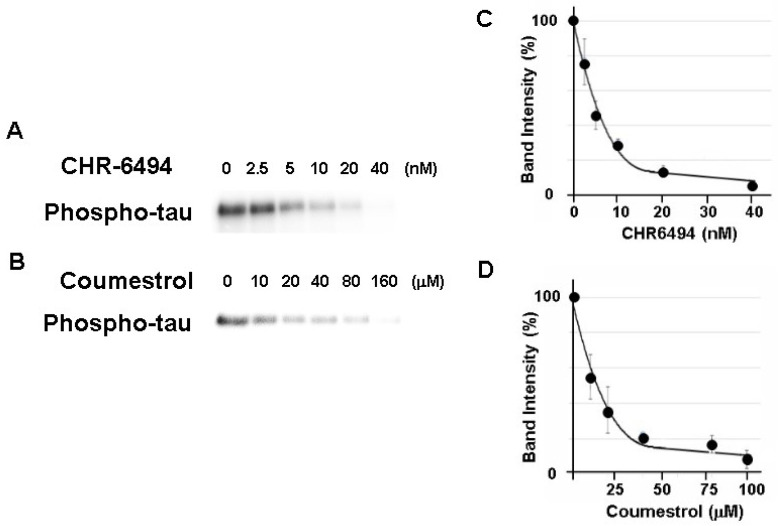
HASPIN kinase assay using the recombinant human tau protein. Recombinant human tau serves as a HASPIN substrate in vitro. HASPIN inhibitor CHR-6494 (**A**,**C**) and coumestrol (**B**,**D**) suppressed phosphorylation of the tau protein. IC50 values were determined according to the effect of different concentrations of the HASPIN inhibitors (**C**,**D**). Data are presented at the mean ± SD of three independent experiments.

**Figure 2 biology-12-00320-f002:**
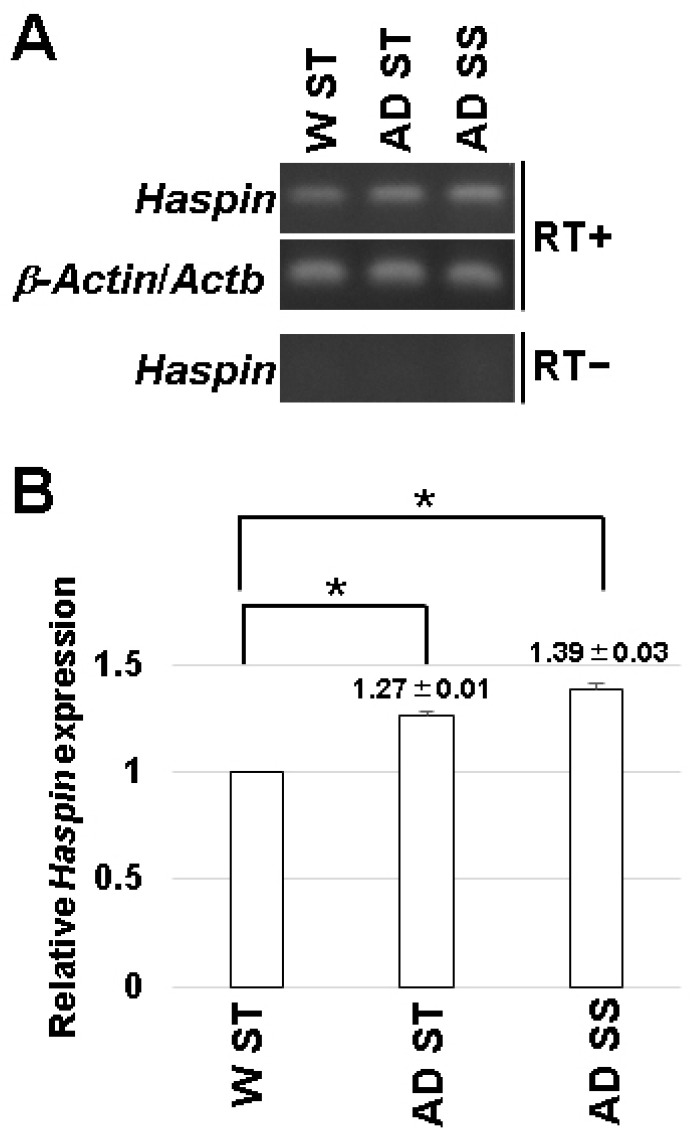
*HASPIN* expression determined in the hippocampus by RT-PCR analysis. WT ST: wild-type mice fed control food, AD ST: 5xFAD mice fed control food, and AD SS: mice fed food including soybean sprouts. Without reverse transcriptase (RT−), the amplified products of *HASPIN* were not detected in each lane. *Actb/β-actin* was used as an internal control (**A**). Semi-quantitative RT-PCR results from the gels using the ImageJ software (**B**). The data are expressed as the mean ± standard error (SE). *, *p* < 0.01.

**Figure 3 biology-12-00320-f003:**
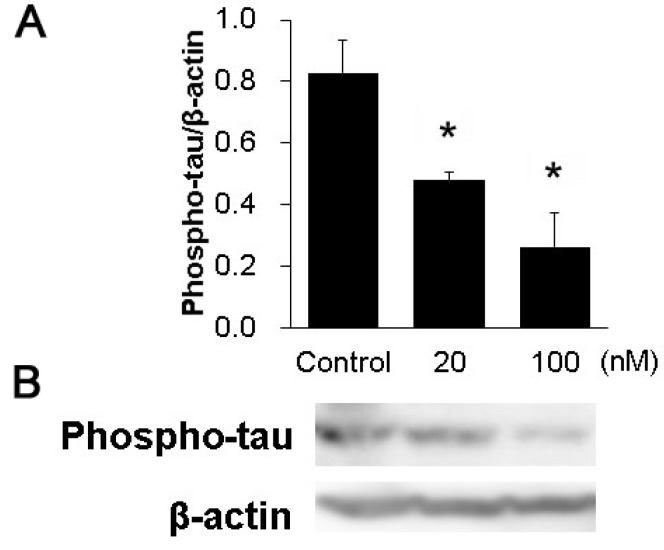
Effect of the HASPIN inhibitor CHR-6494 on tau phosphorylation in the hippocampal slices: Quantitative analysis of tau phosphorylation (**A**). Immunoblot of tau phosphorylation with CHR-6494 in the hippocampal slices (**B**). The data are expressed as mean ± SE. *, *p* < 0.01 vs. control group.

**Figure 4 biology-12-00320-f004:**
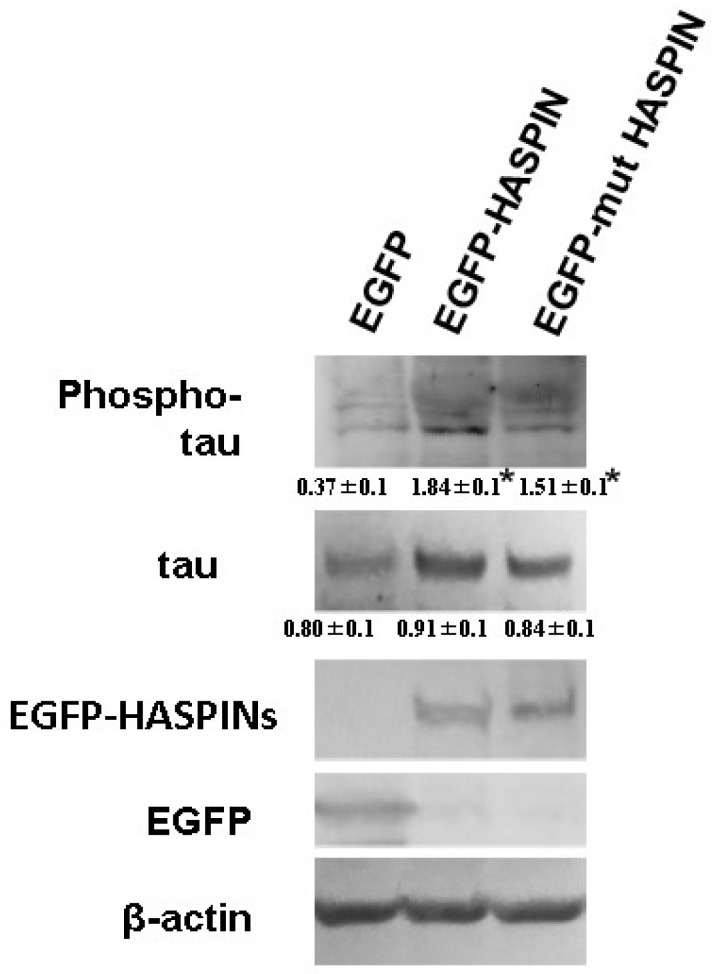
Phosphorylation of the tau protein in HT22 cells of the mouse hippocampal neuronal cell line. The *egfp*-*HASPIN* and mutant *egfp*-mutant *HASPIN* were transfected into HT-22 cells. The amount of phosphorylated tau was higher in wild-type or mutant HASPIN-expressed HT-22 cells than in EGPF-re-expressed HT-22 cells. Tau phosphorylation was more intense in cells expressing wild-type HASPIN. The numbers in the figure indicate the signal intensity relative to β-actin in each lane. The data are expressed as the mean ± SD. *, *p* < 0.05 vs. control group.

**Figure 5 biology-12-00320-f005:**
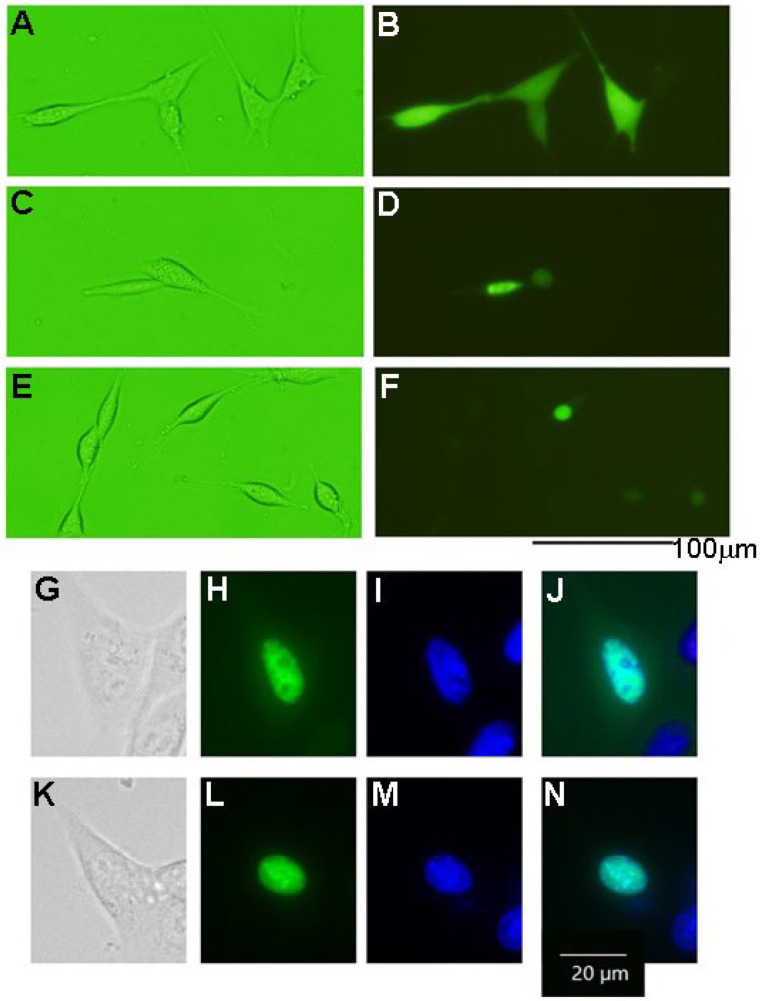
Localization of HASPINs in the HT22 cells. (**A**,**B**) indicate EGFP-expressing cells; (**C**,**D**,**G**–**J**) indicate EGFP-HASPIN expressing cells; (**E**,**F**,**K**–**N**) indicate EGFP-mutant HASPIN expressing cells. EGFP-expressing cells (**B**), where the fluorescence was localized on the nuclei in the EGFP-HASPIN-expressed cells (**D**,**F**,**H**,**L**). (**G**–**N**) show enlarged views of cells. (**I**,**M**) indicate nuclei with Hoechst 33342. (**J**,**N**) indicate the merging of (**H**,**I**) and (**L**,**M**), respectively.

**Figure 6 biology-12-00320-f006:**
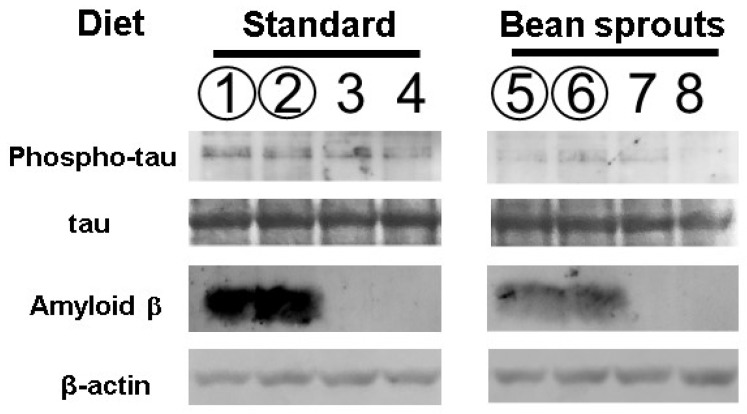
Phosphorylation of the tau protein and the amount of amyloid β of the 5xFAD fed a diet containing soybean sprouts. The number of proteins in mice hippocampi were examined by Western blot analysis. Lanes 1, 2, 5, and 6 indicate the hippocampal protein of 5xFAD mice. Lanes 3, 4, 7, and 8 indicate the hippocampal protein of wild-type mice. The odd lanes are males, and the even lanes are females. Experiments with different individuals were performed three times. Differences between each sample were determined using the two-tailed paired *t*-test, and a *p*-value < 0.05 was considered statistically significant. The signals of the phosphorylated tau and amyloid β in mice fed soybean sprouts were significantly weaker in the wild-type and 5xFAD mice.

**Figure 7 biology-12-00320-f007:**
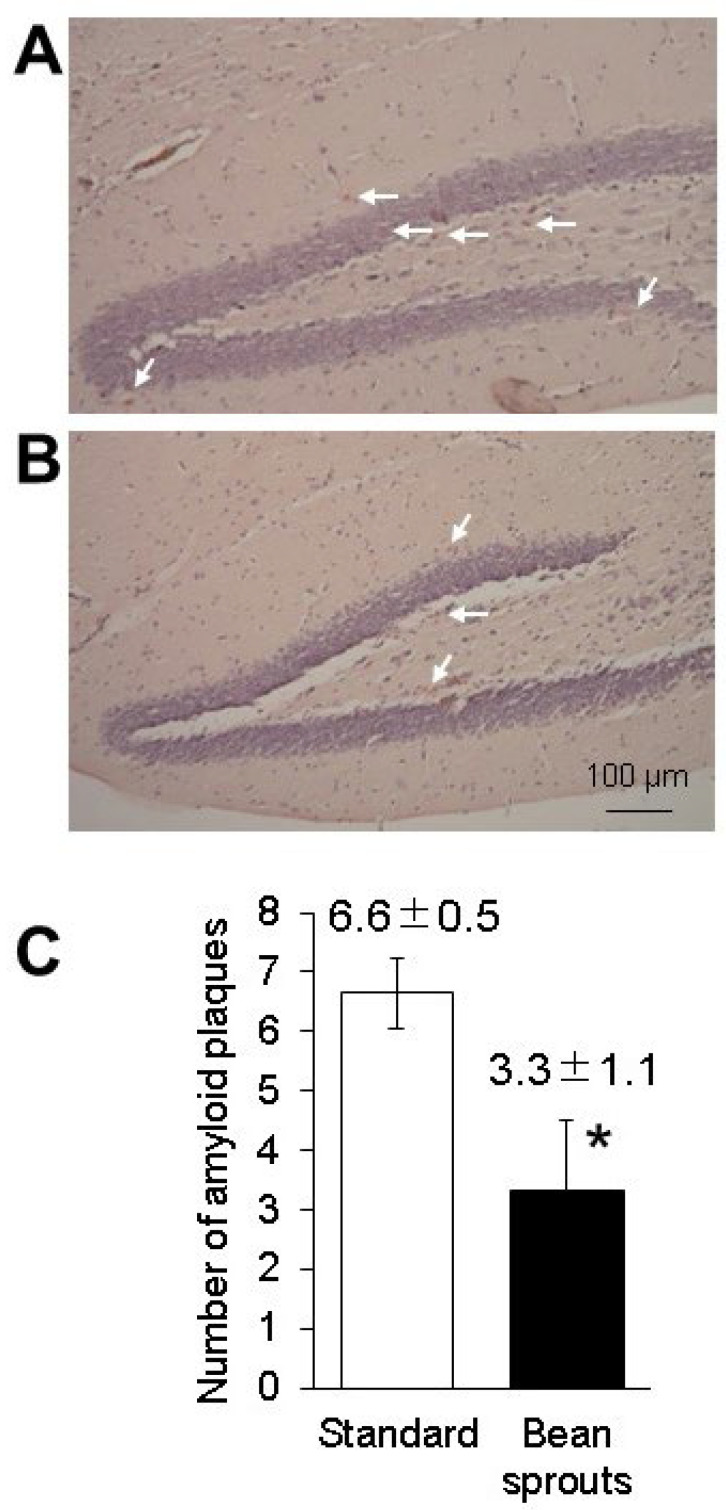
Quantification of the amyloid plaque pathology in the hippocampus of 5xFAD mice fed a diet containing soybean sprouts. Photographs of hippocampal sections from 5xFAD mice fed a standard diet (**A**) and a diet including soybean sprouts (**B**); the average number of amyloid plaques was quantified on Congo red-stained brain slices (**C**). *, *p* < 0.05 vs. 5xFAD mice fed standard diet. Arrows indicate plaques. Data are expressed as the mean ± SD.

**Figure 8 biology-12-00320-f008:**
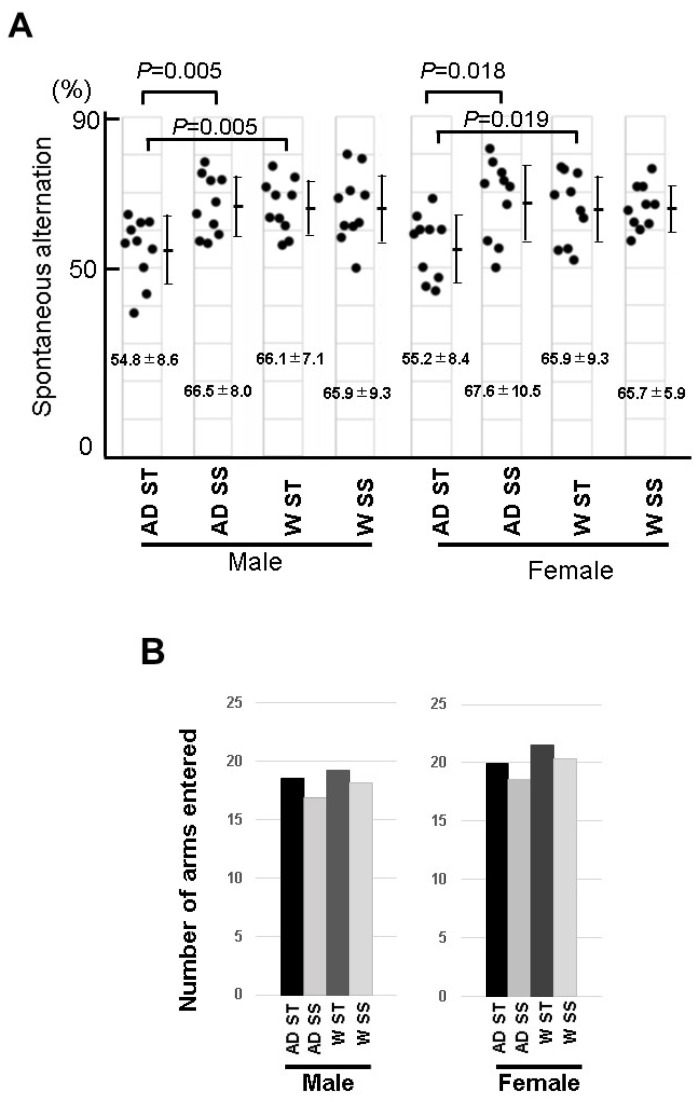
Analysis of the spatial working memory in the Y-maze test of the 5xFAD mice fed a diet containing soybean sprouts: (**A**) five-month-old wild-type (W) and 5xFAD (AD) mice were fed a diet containing soybean sprouts (SS) or a standard diet (ST) for three months, and their behavior was observed. The male and female 5xFAD mice showed poor alternation performance in the Y-maze test, but the wild-type mice and 5xFAD mice fed a diet containing soybean sprouts presented normal performance. (**B**) Total number of arm entries in the Y-maze test. The number of entries into each arm between the soybean sprout-fed and standard-fed mice did not significantly differ between the wild-type and 5xFAD mice. The data are expressed as the mean ± SD.

**Table 1 biology-12-00320-t001:** Antibodies used in the current study.

Name	Reference (Product Code)
anti-tau-mouse IgG	Abcam (ab80579)
anti-phosphorylated tau-rabbit IgG	Cell Signaling Technology (#29957)
anti-β-amyloid rabbit IgG	Cell Signaling Technology (#51374)
anti-EGFP monoclonal antibody	Tanaka et al. [1]
anti-β-actin mouse IgG	Proteintech (HRP-60008)
anti-rabbit IgG HRP-linked whole Ab (from donkey)	Amersham Biosciences (NA934V)
rabbit anti-rat IgG HRP-secondly antibody	DAKO (P0450)
anti-rabbit IgG HRP-linked secondary antibody	Cell Signaling Technology (#7074)
anti-mouse IgG HRP-secondly antibody	Cell Signaling Technology (#7076)

## Data Availability

The datasets used and analysed during this study are available from the corresponding author on reasonable request.

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
