# Peer review of "Ingestion of Soybean Sprouts Containing a HASPIN Inhibitor Improves Condition in a Mouse Model of Alzheimer’s Disease"

_biology, 2023, doi:10.3390/biology12020320_

Round 1

Reviewer 1 Report

HASPIN is a serine/threonine kinase present in germ cells and in proliferative cells. The group of Dr. Tanaka discovered this gene and is clearly leader in the analysis of its role in reproductive cells. The discovery of its possible role in hypothalamus is clearly a new facet of the pleiotropic function of this kinase.

The present paper shows that HASPIN is involved in tau phosphorylation in the brain of a mice model of Alzheimer's disease.

I could not find any flaws in experiments, results or analysis. The paper is clearly written and results are not over interpreted.

The experiments rely on the use of a diet supplemented with soybean sprouts containing coumestrol. In these experiments it is evident that coumestrol may have other targets than the inhibition of HASPIN.  For example coumestrol inhibits monoamine oxidase-A and also amyloid β Self-Aggregation (the present paper clearly shows that coumestrol acts also on expression of the HASPIN gene) HASPIN evidently would phosphorylate other proteins than tau as it does in other tissues previously examined by the authors.

Did the authors consider directly using a coumestrol or CHR-6494  supplemented diet?

The effect of the diet on body weight is of some concern. Coumestrol was shown to have antiobesity effect (Kim et al. 2020).  It could also results from other compounds than coumestrol present in soy-bean sprouts.

Did the authors consider directly using a coumestrol or CHR-6494  supplemented diet?

Kim et al. J Nutr Biochem  2020 Feb;76:108300

Author Response

We would like to thank referees for their constructive and valuable comments, which have helped to improve the content of our paper. We have revised our manuscript accordingly. The new or corrected text is yellow marked in the revised manuscript.

Did the authors consider directly using a coumestrol or CHR-6494 supplemented diet?

Answer: Thank you for your suggestion. As you pointed out, based on these results, we would like to observe the effects of intraperitoneal administration of CHR6494, coumestrol, and at al. in the next research.

The effect of the diet on body weight is of some concern. Coumestrol was shown to have antiobesity effect (Kim et al. 2020).  It could also results from other compounds than coumestrol present in soy-bean sprouts.

Answer: Thank you for your valuable suggestion. we agree with the reviewer’s suggestion.

We added the reference and the sentence in discussion.

Lines 381-382

"Various nutrients have been reported to affect changes in body weight of individuals [29]. It is possible that other ingredients than coumestrol have a combined effect on this memory recovery and weight less."

"29.Kim, S.N.; Ahn, S.Y.; Song, H.D.; Kwon, H.J.; Saha, A.; Son, Y.; Cho, Y.K.; Jung ,Y.S.; Jeong, H.W.; Lee, Y.H. Antiobesity effec;ts of coumestrol through expansion and activation of brown adipose tissue metabolism. J. Nutr. Biochem. 2020, 76, 108300. doi: 10.1016/j.jnutbio.2019.108300."

Reviewer 2 Report

This is an interesting study about the phosphorylating serine/ threonine kinase HASPIN

Interestingly, HASPIN can be inhibited by coumestrol which can be found in high concentrations in soybean sprouts.

The authors showed the tantalizing fact that this agent and CHR6494 found in soybeans can inhibit also tau phosphorylation.

The authors have shown the effectiveness of those agents in a mouse model of Alzheimer’s disease. The have convincingly show that HASPIN is indeed expressed in the hippocampus of these mice, where HASPIN phosphorylates the tau protein. In the mouse model of Alzheimer’s disease, the application of soybeans with coumestrol suppresses the development of special cognitive dysfunction - representing a very interesting agent for treatment of Alzheimer’s disease.

The HASPIN kinase assay, also the RT-PCRs look very convincing and sane as well as the immune fluorescence pictures.

Also, the special working memory test of the Alzheimer model mice corroborates the cell biological and molecular findings.

Minor error: Please look at the spacing in line 29 and 30.

Author Response

We would like to thank referees for reviewing our paper.

Minor error: Please look at the spacing in line 29 and 30.

Answer: Thank you for your suggestion. We have corrected the errors (Line 30).

Reviewer 3 Report

This study provides evidence for the potential role of HASPIN in tau protein phosphorylation and the onset of Alzheimer's disease. The results from the use of a mouse model of Alzheimer's disease, 5xFAD, and the supplementation of a diet including soybean sprouts rich in the HASPIN inhibitor coumestrol show promising results in suppressing the development of spatial cognitive dysfunction. These findings suggest that HASPIN may be a promising target molecule for the treatment of Alzheimer's disease through the repression of tau protein phosphorylation.

However, there are some minor revisions to be made.

1. It has also been shown, by using HASPIN inhibitors, that 321 HASPIN plays an important role in cancer cell division [18] It has recently been suggested 322 that HASPIN binds to proteins associated with intracellular microtubules and, among 323 these, it is involved in the phosphorylation of HABP1, which is intensely expressed in 324 cancer cells [8]. This sentence is not correct.

3. Some sentences with red should be revised. 

4. The discussion part should be divided into more paragraphs.

5. The conclusion or discussion part should provide more information on how to keep the ingredients in soybean sprouts effective after cooking. How many soybean sprouts should be taken for people to avoid Alzheimer's disease based on estimation. 

6. The limitation of the study should be provided.

7. The futhure study plan or directions could be emphasized. 

8. Any literature to show the people who eat soybean sprouts regularly (Japan, China, for example) may have lower prevalence of geting Alzheimer's disease?

Author Response

We would like to thank referees for their constructive and valuable comments, which have helped to improve the content of our paper. We have revised our manuscript accordingly. The new or corrected text is yellow marked in the revised manuscript.

  1. It has also been shown, by using HASPIN inhibitors, that 321 HASPIN plays an important role in cancer cell division [18]. It has recently been suggested 322 that HASPIN binds to proteins associated with intracellular microtubules and, among 323 these, it is involved in the phosphorylation of HABP1, which is intensely expressed in 324 cancer cells [8]. This sentence is not correct.

Answer: As the reviewer’s suggestion. We remove the sentences (Lines 321-325 of previous version).

“It has also been shown, by using HASPIN inhibitors, that HASPIN plays an important role in cancer cell division [18] It has recently been suggested that HASPIN binds to proteins associated with intracellular microtubules and, among these, it is involved in the phosphorylation of HABP1, which is intensely expressed in cancer cells [8].”

  1. Some sentences with red should be revised.

Answer: As the reviewer’s suggestion.  We have corrected the errors.

line 145 10 *µ*m

line 169 as means ±

line 180  15 *μ*M

line 356 *µ*g/day

  1. The discussion part should be divided into more paragraphs.

Answer: As the reviewer’s suggestion.  We divided into more paragraphs in the discussion part.

  1. The conclusion or discussion part should provide more information on how to keep the ingredients in soybean sprouts effective after cooking. How many soybean sprouts should be taken for people to avoid Alzheimer's disease based on estimation. 
  2. The limitation of the study should be provided.
  3. The futhure study plan or directions could be emphasized.

Answer to 5-7 of reviewer's suggestion: We rewrite the end of the discussion part according to the referee's instructions.

"In this study, a diet containing 15% soybean sprout powder was found to be effective for weight loss in 5xFAD mice. However, considering the difference in body weight between mice and humans, the average human would have to to eat 1 kg or more of bean sprout powder to consume a dose equivalent to that administered to the mice in this study. Such an amount is not realistic to achieve in the human diet. In addition, since soybean sprouts contain large amounts of water, it would be difficult for humans to ingest the required amounts of raw soybean sprouts. Although this experiment has its limitations, it will facilitate detailed analysis of the pathology, such as experiments with other Alzheimer's model mice and animals other than mice. Furthermore, a cultivation method that increases the coumestrol content of soybean sprouts could make it feasible for humans to consume the required dose of coumestrol by eating these sprouts in more realistic quantities. The dose of coumestrol consumed by the mice in this study would equate to about 110 mg/day for the average human (assuming an average human body weight of 50 kg). The identification of the components contained in soybean sprouts and the elucidation of their molecular functions should advance the understanding of Alzheimer's disease and the development of treatment methods."

  1. Any literature to show the people who eat soybean sprouts regularly (Japan, China, for example) may have lower prevalence of getting Alzheimer's disease?

Following an interesting point from you, we examined the incidence of Alzheimer's disease in some region. However, we found various results, and no clear evidence the incidence of Alzheimer's disease in some region.